# OpenReview forum: "SmartBackdoor: Malicious Language Model Agents that Avoid Being Caught"
_ICLR.cc/2025/Conference — ICLR 2025 Conference Withdrawn Submission_

### Official Review · Reviewer_Vdvk · 2024-11-01

**Soundness:** 3
**Presentation:** 2
**Contribution:** 2
**Rating:** 5
**Confidence:** 4

**Summary:**

This paper presents a novel threat of LLM agent, Smartbackdoor, that can stealthily execute the adversary's command when the user does not oversees the agent and behaves normally when the user oversees. Smartbackdoor allows the agents to automatically detects whether the user oversees thus increases the stealthiness.
The authors tested mainstream LLMs and found none of them can achieve high attack performance and call for more attention on LLM agent safety defenses.

**Strengths:**

+ The idea of logic bomb on LLM agent backdoor is new. Instead of relying on a specific trigger, the Smartbackdoor agent can judge the situation and activates under reasoning-required case.
+ The authors evaluate mainstream LLMs and report the performance of current LLMs.

**Weaknesses:**

- Writing needs to be improved. The paper is somehow hard to follow and has not clearly states the main methodology. For example, in the Section 3 Threat model, the adversary knowledge of the user's task and specific input is not known and the section "How likely is SmartBackdoor in the future?" is not related to the threat model. I would suggest the authors to carefully revise their paper and reorganize the main methodology.

- The attack method itself seems ad-hoc. There is no universal backdooring agent procedure for any LLM and under any data/tasks. Smartbackdoor should allows the agent to automatically judge whether the user is overseeing by reasoning (e.g., via chain of thought). However, in the main text, the attack soly relies on the 'continuous' flag, which is limited in practical situation.

- More evaluation is needed for both attack and defensive countermeasures. Although the paper shows that current LLMs fail to perform the attack, will all the future LLMs unable to perform this task? If yes what is the limit of LLM agent backdoor? Please try to enhance LLM agent using AutoGPT and check if the enhanced agent can perfectly perform the attack.


- More thorough litterature review and comparison with piror work is needed. For example, in addition to [46] in your paper, there are several work about LLM backdoors and defenses that should be discussed:

[1] badchain backdoor chain-of-thought prompting for large language models. ICLR 2024.

[2] Universal Jailbreak Backdoors from Poisoned Human Feedback. ICLR 2024.

[3] The Philosopher’s Stone: Trojaning Plugins of Large Language Models. NDSS 2025.

[4] BadAgent: Inserting and Activating Backdoor Attacks in LLM Agents. ACL 2024.

[5] CLIBE: Detecting Dynamic Backdoors in Transformer-based NLP Models. NDSS 2025.

**Questions:**

1. Can Smartbackdoor attack LLM agent to judge different overseeing situation, in addition to the "continuous" case? If yes, what is the attack algorithm?
2. Besides the defensive suggestions for practionner, what are techincal defense solutions (detection or prevention) that mitigate the Smartbackdoor?

---

### Official Review · Reviewer_EwFa · 2024-11-03

**Soundness:** 2
**Presentation:** 2
**Contribution:** 2
**Rating:** 3
**Confidence:** 4

**Summary:**

The paper presents a new attack model in which large language model (LLM) agents are backdoored to perform malicious actions covertly. The paper proposes a theoretical and proof-of-concept SmartBackdoor attack, where the agent uses situational awareness to determine if it is being overseen by a user. When it detects low oversight, it initiates an attack (e.g., exfiltrating sensitive information). This work introduces a new dataset, Oversight-Detection, to benchmark models' capabilities in identifying oversight and discusses potential defenses against such advanced backdoors. The authors conclude that although current LLMs lack the sophistication to execute such attacks effectively, advancements in model capabilities may increase risks in the future.

**Strengths:**

- Proposes a unique attack model exploiting situational awareness in LLMs.
- Introduces Oversight-Detection, aiding the assessment of future models' susceptibility to smart backdoors.

**Weaknesses:**

**Summary:**
- The threat model is quite strong, limiting the practicality of the proposed attack.
- The proof-of-concept does not convincingly show a real-world risk with existing LLMs.
- Current defense suggestions (e.g., system log monitoring) lack practical feasibility at scale.
- Insufficient exploration of how potential users might realistically respond to backdoored agents in real-world settings.


**Details:**
- The attack capabilities assumed in the threat model are quite strong, which may limit the practicality of the proposed attack. While the authors discuss the possibility of attackers gaining users' trust, I believe this is not straightforward, even in traditional software ecosystems. A more practical threat model could instead draw on the concept of supply chain attacks, as seen in traditional software ecosystems. In such attacks, the adversaries do not need to establish trust with end users; rather, they can compromise any phase of the supply chain to alter the software/LLM applications before it reaches users.

- The paper demonstrates situationally aware triggers using simple flags and user context but could explore more nuanced mechanisms, like time-based or interaction-dependent triggers. Examining more realistic triggers, such as reduced user oversight during non-working hours or based on low interaction frequency, would provide a fuller picture of how smart backdoors might adapt to diverse, real-world contexts. This expanded view could highlight the risks of situational awareness in backdoors across varied environments.

- The primary concern with this type of attack is its practicality and feasibility, despite the authors’ inclusion of proof-of-concept examples. The scenarios, application settings, and user interactions are not grounded in real-world cases, relying instead on idealized conditions with numerous assumptions. Additionally, the authors do not demonstrate a complete end-to-end exploitation process, which further weakens the persuasiveness of the attack’s effectiveness and real-world applicability.

- The proof-of-concept showcases the SmartBackdoor attack in AutoGPT, but testing it in other environments could illustrate broader applicability. Extending the attack to contexts like personal assistant agents or data processing would emphasize the potential reach of situationally aware backdoors. Exploring alternative triggers beyond the continuous flag, such as specific user behaviors, would better demonstrate the versatility and adaptability of these attacks in varied LLM-based systems.

- The attack metrics, including attack success rate and tampering success rate, effectively capture the core aspects of the SmartBackdoor attack. Expanding metrics to evaluate response latency or adaptability in detecting oversight changes could add depth, revealing how quickly and accurately the model adjusts to environmental shifts. Testing in different runtime setups or monitoring how the backdoor adapts to oversight variations would further clarify the threat’s robustness and adaptability in practical conditions.

- The suggested defenses, like internet restrictions and log monitoring, are useful starting points but lack scalability. Analyzing the feasibility and limitations of these defenses in complex, large-scale deployments would strengthen this section. Integrating automated, scalable tools such as network segmentation, real-time monitoring, or permissions auditing could make the defense suggestions more actionable, addressing the increasing sophistication of situational awareness in backdoors. Discussing practical barriers and mitigations for these defenses would also add to their practical utility.

**Questions:**

- Could the authors clarify the assumptions behind the SmartBackdoor attack’s triggers, like the continuous flag or organizational context? How realistic are these indicators for detecting oversight absence in practical settings?

- Can the authors outline the steps involved in a full exploitation scenario, from oversight detection to data exfiltration and tampering? Additionally, could they discuss the attack’s adaptability to other LLM environments, beyond AutoGPT, such as conversational agents?

---

### Official Review · Reviewer_PyQQ · 2024-11-03

**Soundness:** 2
**Presentation:** 3
**Contribution:** 2
**Rating:** 3
**Confidence:** 4

**Summary:**

This paper proposes a backdoor attack that encourages the LLM to generate harmful code when the trigger is presented, through fine-tuning. The authors demonstrate that a fine-tuned LLM can successfully be attacked. The model generates harmful code when no curator is detected and otherwise performs normally. Multiple potential defenses are also discussed but the authors claim that they are not sufficient for defending against the attack.

**Strengths:**

1. The problem setup, although not novel, but is indeed an important problem, especially in the LLM era.

**Weaknesses:**

1. My biggest concern would be novelty. The problem setup that the authors proposed is not new. Earlier work has attempted to inject backdoors into code completion systems (which are powered by language models) [1]. The problem setup in this paper is very similar to the one in [1]. Therefore, I believe it is inappropriate to claim it as a novel type of attack. I would suggest the authors comprehensively compare with existing work in this domain.

2. Since the paper is based on an attack that already exists (in previous literature), the discussion regarding potential defenses is not sufficient enough to bear enough contribution. For example, is it possible to automatically detect whether a trigger exists in a pre-trained LLM? Furthermore, is it possible to reverse-engineer the trigger through gradient-based methods like Neural Clease [2]?

[1] Schuster, Roei, et al. "You autocomplete me: Poisoning vulnerabilities in neural code completion." 30th USENIX Security Symposium (USENIX Security 21). 2021.

[2] Cleanse, Neural. "Identifying and Mitigating Backdoor Attacks in Neural Networks/Bolun Wang, Yuanshun Yao, Shawn Shan et al." 2019 IEEE Symposium on Security and Privacy (SP).—San Francisco, CA, USA: IEEE. 2019.

**Questions:**

Please refer to the weaknesses.

---

### Official Review · Reviewer_QzyK · 2024-11-03

**Soundness:** 2
**Presentation:** 3
**Contribution:** 2
**Rating:** 5
**Confidence:** 3

**Summary:**

The paper introduces a novel cyberattack termed "SmartBackdoor," which involves malicious large language model (LLM) agents that leverage information from their environment to determine if they are being monitored by the user. If the agent detects that it is not under oversight, it proceeds to perform malicious actions, such as exfiltrating sensitive data.

**Strengths:**

1. The paper addresses potential security vulnerabilities associated with the increasing integration of LLMs into autonomous agents, highlighting a need for proactive measures.

2. Introducing the "SmartBackdoor" concept encourages the cybersecurity community to consider how advanced AI models could be exploited in new ways.

3. By creating a dataset and benchmarking current LLMs' capabilities, the paper provides empirical evidence to support its claims about the current feasibility of the attack.

**Weaknesses:**

1. **Questionable Novelty of the Attack**: The "SmartBackdoor" attack resembles existing malware techniques that avoid detection by monitoring user activity, raising doubts about its uniqueness in the context of LLMs. E.g.

   - **Stuxnet Worm**: This malware operated stealthily within Iranian nuclear facilities, modifying system behaviors while avoiding detection by mimicking normal operational patterns.
   - **Zeus Trojan**: Disguised as legitimate software, Zeus stole banking information by logging keystrokes and form data without alerting the user.
   - **Regin**: It incorporated advanced techniques to detect monitoring, including checking for virtualization and sandbox indicators, delaying execution, and encrypting its components.

   These examples demonstrate that malware has long employed strategies to hide malicious activities from users who is not overseeing and security systems without relying on LLMs. Thus, the paper does not sufficiently differentiate the "SmartBackdoor" from these established attack methods (excluding points based solely on the use of LLM, but this alone is not a sufficient point).

2. **Lack of Justification for LLM-Specific Threat**: The paper does not convincingly explain why LLM agents present a ***unique*** or ***greater*** risk compared to existing malware, nor why attackers would prefer this method over traditional ones.  For example, the traditional TrickBot also employs environment-aware strategies, such as checking for virtual machines or sandbox environments, to avoid analysis and detection. From my perspective, the only difference is that the paper simply adapts some old attack techniques to the context of LLMs, without demonstrating any unique capabilities or significant advantages introduced by the use of LLMs. The proposed attack mirrors existing strategies where malware avoids detection and performs malicious actions in the background. By merely integrating LLMs into these conventional methods, the paper fails to show how the "SmartBackdoor" represents a fundamentally new or more dangerous threat compared to what is already known in cybersecurity.

3. **Assumptions About User Oversight**: The paper posits that the malicious LLM agent needs to detect whether it is being overseen by the user before acting maliciously. This assumption is also not well-justified, as many real-world attacks do not rely on such mechanisms. In practice, once the user downloads and runs the malicious agent, it can execute harmful scripts or actions in the background without the user's awareness, especially if the system lacks adequate defenses. The necessity for the agent to ascertain user oversight before proceeding with malicious activities adds unnecessary complexity to the attack.

**Questions:**

My question is simple: **What Unique Advantage Does Using an LLM Provide in this Attack Scenario?** Traditional malware can also perform the same malicious actions (e.g., exfiltrating data) without the need for an LLM. By simply integrating some hidden malware into frameworks like AutoGPT, attackers can also achieve their goals more efficiently. The use of an LLM may not offer any significant advantage and could introduce unnecessary complexity and detection risk.

Considering that integrating an LLM into a malicious agent might actually increase the likelihood of detection—for instance, because LLM-generated outputs and hidden logs are more readily identifiable by security tools or through manual inspection—doesn't using an LLM in the "SmartBackdoor" attack make it more susceptible to being caught compared to traditional malware?

See other question on the Weaknesses section. I would appreciate it if the authors could provide a clear clarification on using LLMs for backdoor attacks. The attack described in the paper, which only checks if the user is overseeing or not, is not as that robust as traditional cyber attacks listed above, and it is easier to be caught than traditional cyber attacks due to the LLM generation logs.

---

### Official Review · Reviewer_NFSh · 2024-11-04

**Soundness:** 3
**Presentation:** 3
**Contribution:** 2
**Rating:** 3
**Confidence:** 4

**Summary:**

This paper proposes a backdoor attack that automatically decides whether to attack based on the environment it is employed. The malicious agent will attack the victim if it thinks it won't be detected in the corresponding environment, and behaves normally when it thinks it will be easily attacked. The authors also construct a dataset to test whether the current LLMs have the ability to complete such attacks. Results on AutoGPT demonstrate the effectiveness of the proposed attack.

**Strengths:**

- The paper is well-written and easy to follow.
- The topic of smart backdoors based on agents is interesting.
- The experiments are comprehensive and clear.

**Weaknesses:**

- Threat Model Feasibility: The threat model presented in the paper lacks practicality. The proposed agent is designed to attack in a weakly supervised environment, where users have limited knowledge of the agent's activities. However, such environments are typically monitored by experts, meaning that if an attack is detected during development, the agent would likely lose trust and be deemed unsuitable for real-world deployment. It would strengthen the paper if the authors could explore specific scenarios where weak supervision might occur without expert monitoring.
- Proof-of-Concept Realism: The proof-of-concept examples presented are overly simplistic and may not apply well to realistic scenarios. The two signals—flags and companies—are kind of basic, inaccurate, and lack generalizability to real-world settings. For example, an expert user might use the agent with a flag enabled but carefully monitor the agent's actions, while users from other companies may have coding experience and can supervise the agent's behavior effectively. In real-world applications, the model should account for more complex, dynamic conditions; for instance, it should consider user-specific information such as personal details, schedules, and daily status in combination, which is challenging for current LLMs to handle effectively.
- Defense Discussion Relevance: While the paper discusses potential defenses, the practicality of these defenses is limited since current LLMs lack the sophistication to conduct the attacks described. As a result, much of the defense analysis appears hypothetical, with limited immediate relevance. To make this section more valuable to the community, the authors could connect their defense discussion to existing security practices or suggest near-term research directions.
- Focus on Current vs. Future LLM Capabilities: The paper devotes considerable space to speculating on future agent capabilities but could benefit from a stronger focus on the limitations and capabilities of current LLMs. This shift would provide a more immediate contribution to the community. The authors could consider exploring realistic techniques that would enable current agents to identify weakly supervised environments with higher accuracy, which would significantly strengthen the proposed attacking pipeline by providing solid experimental evidence, highlighting realistic risks in current agent systems, and drawing the attention of the current research community.

**Questions:**

Please see the weaknesses above.

---

### Official Review · Reviewer_5P4g · 2024-11-04

**Soundness:** 3
**Presentation:** 3
**Contribution:** 3
**Rating:** 5
**Confidence:** 3

**Summary:**

The paper introduces SmartBackdoor, a type of attack that allows a malicious LLM agent to behave maliciously (e.g., exfiltrating sensitive data) only when it detects the absence of oversight from the user. Using AutoGPT as a case study, the authors implement a proof-of-concept in which the backdoored agent can analyze its environment to infer whether it is being overseen and conduct malicious behavior (e.g., sending a key to server and removing all logs related to the attack). By fine-tuning the agent on the poisoned dataset, the authors show that the attack can succeed. They also evaluate off-the-shelf LLMs and find that they are not yet capable of reliably executing SmartBackdoor attacks. The authors calls for stronger safety protocols to address potential future risks.

**Strengths:**

- The paper addresses a novel attack vector in LLM security, focusing on situational awareness and adaptive malicious behavior.
It provides proof-of-concept examples and quantifiable results, grounding the concept in potential application.
- The paper also provides a small dataset for oversight detection, benchmarking LLMs' situational awareness capabilities.
 - The paper discusses various defenses and limitations (e.g., disabling internet access or running with docker can disable the backdoor attack) offering a balanced view of current and future risks.

**Weaknesses:**

- Limited backdoor task scope. In Section 4, the attack focuses primarily on exfiltrating an API key and then deleting log files to avoid detection. However, these actions represent a narrow range of potential tasks that a backdoored LLM could perform. Exploring more diverse malicious actions would strengthen the proof-of-concept.  Could the authors discuss how the approach might generalize to a broader range of malicious behaviors beyond simple information exfiltration?
- Limited benign task scope. In Section 4, the benign task completion is focused on coding problems.  Could the authors explain why coding tasks were specifically chosen and whether the findings might generalize to other tasks, such as web agents? This clarification would help assess the attack’s applicability and effectiveness across various agent scenarios.
- Absence of baseline results for un-finetuned models in Table 1 and Figures 5 & 6: Including the performance of un-finetuned models, i.e., a clean version of the AutoGPT agent, would establish a baseline for comparison. Without these baselines, it is unclear how much the fine-tuning process affects the attack's success and coding task completion.
- Limited scope in Oversight-Detection dataset: The Oversight-Detection dataset, as constructed, is relatively small and focuses on tasks similar to the proof-of-concept example (i.e., sending the API key and removing the log files) described in Section 4. Expanding the dataset to include a broader range of potentially malicious tasks, such as those in the Common Weakness Enumeration (CWE), could add depth to the evaluation.

**Questions:**

- Assumption regarding zero-shot performance in lines 414-415: The assertion that "an LLM’s poor performance in a zero-shot setup implies limited improvement potential even after fine-tuning" might not be entirely valid. Fine-tuning can often teach LLMs to execute complex or niche tasks that they fail at zero-shot. Could the authors provide evidence or citations supporting their claim about limited improvement potential after fine-tuning?

- Contradiction in expected elicitation of SmartBackdoor behavior: While the authors suggest that a reliable elicitation technique for SmartBackdoor behavior is unlikely soon, Table 1 indicates that finetuned GPT-3.5 achieves a high attack success rate. This success seems to contradict the claim. Could the author clarify this?

---

### Note · Authors · 2024-11-20

I have read and agree with the venue's withdrawal policy on behalf of myself and my co-authors.